# Opioid-related deaths during hospital admissions or shortly after discharge in the United Kingdom: A thematic framework analysis of coroner reports

**Dan Lewer**[1,2]*, **Thomas D. Brothers**[1,3], **Magdalena Harris**[4], **Kirsten L. Rock**[5], **Caroline S. Copeland**[5]

**1** Department of Epidemiology and Public Health, University College London, London, United Kingdom, **2** Bradford Institute for Health Research, Bradford Teaching Hospitals NHS Foundation Trust, Bradford, United Kingdom, **3** Department of Medicine, Dalhousie University, Halifax, Canada, **4** Department of Public Health, Society and Environments, London School of Hygiene & Tropical Medicine, London, United Kingdom, **5** Institute of Pharmaceutical Sciences, Centre for Pharmaceutical Medicine Research, King's College London, London, United Kingdom

\* d.lewer@ucl.ac.uk

**Data Availability Statement:** The full dataset cannot be shared publicly because some coroner

## Abstract

### Background

People who use heroin and other illicit opioids are at high risk of fatal overdose in the days after hospital discharge, but the reasons for this risk have not been studied.

### Methods

We used the National Programme on Substance Abuse Deaths, a database of coroner reports for deaths following psychoactive drug use in England, Wales, and Northern Ireland. We selected reports where the death occurred between 2010 and 2021, an opioid was detected in toxicology testing, the death was related to nonmedical opioid use, and death was either during an acute medical or psychiatric hospital admission or within 14 days after discharge. We used thematic framework analysis of factors that may contribute to the risk of death during hospital admission or after discharge.

### Results

We identified 121 coroners' reports; 42 where a patient died after using drugs during hospital admission, and 79 where death occurred shortly after discharge. The median age at death was 40 (IQR 34–46); 88 (73%) were male; and sedatives additional to opioids were detected at postmortem in 88 cases (73%), most commonly benzodiazepines. In thematic framework analysis, we categorised potential causes of fatal opioid overdose into three areas: (a) hospital policies and actions. Zero-tolerance policies mean that patients conceal drug use and use drugs in unsafe places such as locked bathrooms. Patients may be discharged to locations such as temporary hostels or the street while recovering. Some patients bring their own medicines or illicit opioids due to expectations of low-quality care, including

reports may be considered sensitive. NPSAD is managed by Dr Caroline Copeland at King's College London. Researchers who want more information about using data from NPSAD should contact npsad@sgul.ac.uk or caroline.copeland@kcl.ac.uk.

**Funding:** This study was conducted as part of the NIHR-funded programme 'Improving Hospital Opiate Substitution Therapy (iHOST)' [Ref: NIHR133022], which included funding for DL and MH. DL was also funded by a NIHR Doctoral Research Fellowship [DRF-2018-11-ST2-016]. The views expressed are those of the author(s) and not necessarily those of the NHS, the NIHR or the Department of Health and Social Care. TDB is supported by the Dalhousie University Internal Medicine Research Foundation Fellowship, a Canadian Institutes of Health Research Fellowship [CIHR-FRN#171259], and through the Research in Addiction Medicine Scholars (RAMS) Program (National Institutes of Health/National Institute on Drug Abuse; R25DA033211). The funders had no role in study design, data collection and analysis, decision to publish, or preparation of the manuscript.

**Competing interests:** The authors have declared that no competing interests exist.

undertreated withdrawal or pain; (b) high-risk use of sedatives. People may increase sedative use to manage symptoms of acute illness or a mental health crisis, and some may lose tolerance to opioids during a hospital admission; (c) declining health. Physical health and mobility problems posed barriers to post-discharge treatment for substance use, and some patients had sudden deteriorations in health that may have contributed to respiratory depression.

## Conclusion

Hospital admissions are associated with acute health crises that increase the risk of fatal overdose for patients who use illicit opioids. Hospitals need guidance to help them care for this patient group, particularly in relation to withdrawal management, harm reduction interventions such as take-home naloxone, discharge planning including continuation of opioid agonist therapy during recovery, management of poly-sedative use, and access to palliative care.

## Background

There are well-established time periods of heightened risk for fatal opioid overdose, including release from prison [1,2], discharge from community or inpatient drug treatment [3–5], and the weeks after a non-fatal drug overdose [6]. Recent research has also found an elevated risk after a hospital stay for medical treatment [7–10]. For example, a study in England found that the risk of fatal overdose was increased four-fold in the two days after hospital discharge; and 1 in 14 opioid-related deaths happen in the two weeks after discharge from an acute medical or psychiatric ward [7]. This study also found that people who use illicit opioids who are admitted to hospital for reasons other than a drug overdose appeared to have a similar likelihood of dying from an opioid overdose while admitted to hospital as when they are in the community, despite the immediate availability of emergency care.

The mechanisms behind these risks have not been studied. However, aspects of hospital policy and practice that may harm patients who use illicit drugs have been studied. Together these issues have been called the 'hospital risk environment' [11], which may offer some possible explanations for risk associated with hospital admissions and discharges. Qualitative and quantitative research with people who use illicit opioids suggest that in-hospital drug use is common [12–15], and 'zero tolerance' approaches may lead people to use drugs in riskier ways such as in locked toilet cubicles. Patients report delayed or low doses of opioid agonist therapy (OAT), in part due to conservative or unclear hospital policies in relation to OAT [16,17]. This may lead patients to secretly stockpile drugs including prescribed OAT because they anticipate inadequate pain or withdrawal management [18]. Patients who use illicit drugs also report stigmatising attitudes among staff [19]. Poor pain and withdrawal management and experiences of stigma can lead to patient-directed discharge [20], which is consistently associated with poor health outcomes and readmission [21,22]. Patients who abstain from drugs or do not receive OAT during a hospital stay may lose opioid tolerance [18], increasing the risk on resumption of opioid use.

Existing studies into overdose deaths around the time of a hospital admission [7–10] have used quantitative analyses of national hospital databases, which have limited contextual information about patients, health services, and social conditions. An understanding of these factors could guide preventative strategies. We therefore studied coroners' reports, which include

narrative reports about the circumstances of death. Since these reports vary in terms of format and length, they are sometimes analysed using qualitative methods such as thematic framework analysis [23,24]. Our research question was: what factors contribute to the risk of fatal overdose around the time of a hospital admission?

## Methods

### Dataset: The National Programme on Substance Abuse Deaths (NPSAD)

NPSAD was set up in 1997 to collate coroner reports on deaths that are related to psychoactive drugs. Coroners in England, Wales, and Northern Ireland voluntarily report a death to NPSAD if the death was considered drug-related, the decedent was known to use illicit drugs, or illicit drugs were detected in post-mortem toxicology testing. As an estimate of coverage, the Office for National Statistics reported 22,497 deaths between 2010 and 2021 where an opioid was mentioned on the death certificate [25], and NPSAD includes 11,831 deaths with an opioid identified in toxicology reports over the same period, suggesting approximately half of relevant deaths are included.

Coroners' investigations use a variety of sources, including statements from witnesses, family, and friends; General Practitioner records; reports from first responders such as police and paramedics; hospital reports; substance abuse team reports; and post-mortem and toxicology reports. Coroners submit summary reports to NPSAD, which are transposed into a database with structured fields including age, sex, housing status, employment status, and drugs detected at post-mortem. We did not have access to hospital records beyond information in the coroners' report.

The King's College London Biomedical & Health Sciences, Dentistry, Medicine and Natural & Mathematical Sciences Research Ethics Sub-Committee re-confirmed (August 2022) that analyses of NPSAD do not require ethics review as all subjects are deceased.

### Selection of reports

We included reports where: (a) the date of death was between 1 January 2010 and 1 November 2021 (few deaths were included after May 2021 due to the time taken by coroners to investigate and report deaths); (b) an opioid was identified in toxicology reports; (c) the death was related to nonmedical opioid use, ie. use of illicit or 'diverted' opioids, (d) the death was either during an acute medical or psychiatric hospital admission or within 14 days of discharge. We excluded deaths if: (a) the duration between discharge and death was not known; (b) death was on hospital premises but the person was not admitted, for example if they visited the hospital to use a public toilet; (c) the person was admitted to hospital for treatment after a drug overdose in the community, and died because of that overdose, rather than due to further illicit drug use while in hospital.

We identified relevant deaths following the process shown in Fig 1. We first extracted cases where an opioid was detected in toxicology reports, and either hospital was the place of death or the text strings 'hosp', 'admission', 'discharge', or 'infirmary' appeared in any field. We screened the structured fields to exclude cases unlikely to meet inclusion criteria. Two authors (DL and CC) double-screened a sample (365/2,443; 15%), with agreement of 98.4% and Cohen's kappa of 0.88. We then retrieved the original reports and determined whether they met inclusion criteria.

### Analysis

We used thematic framework analysis [26,27] to identify and explore factors that may contribute to the risk of fatal overdose around the time of a hospital admission. Framework analysis

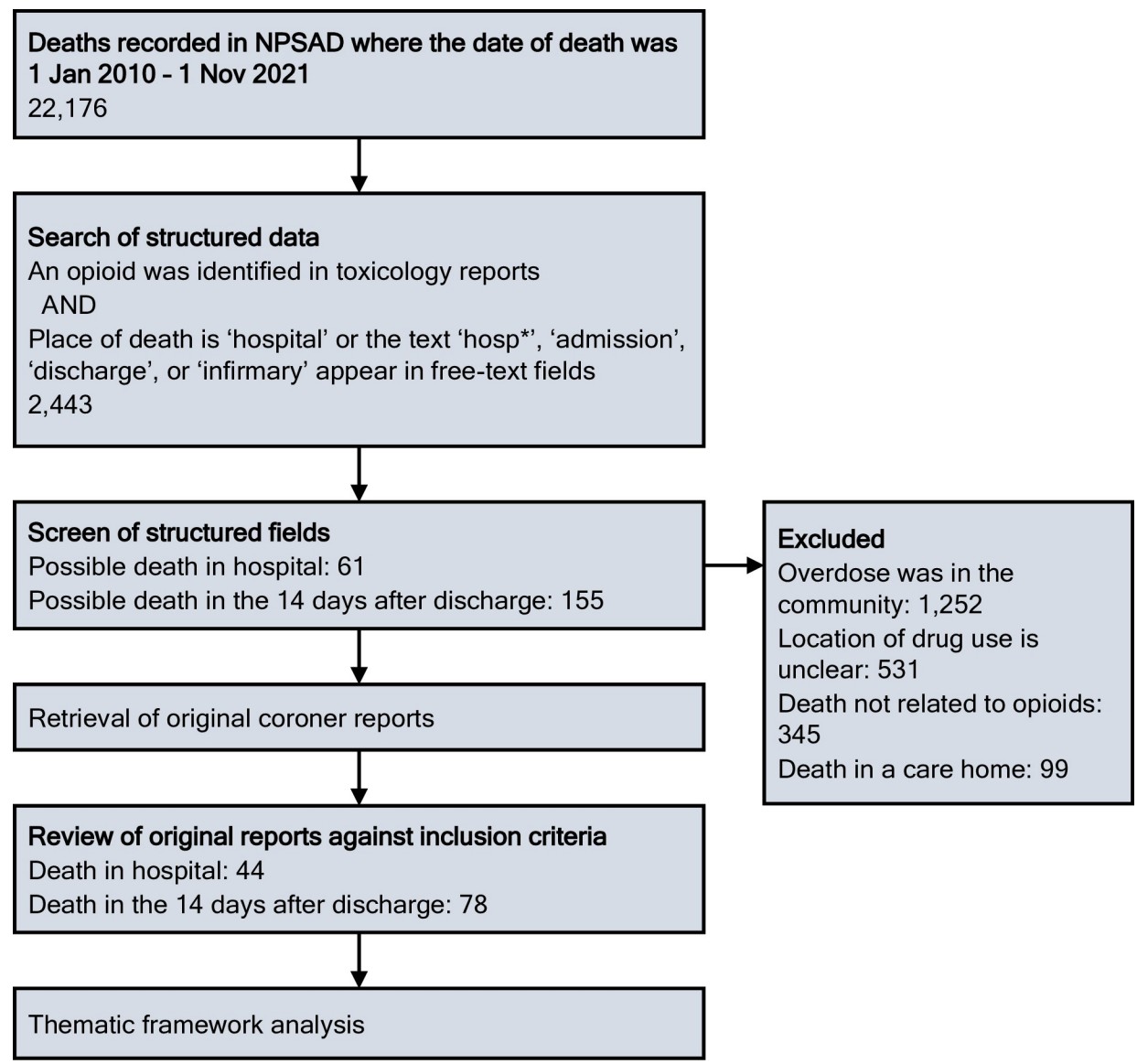

**Fig 1. Flow chart of the process of selecting coroner reports for analysis.** NPSAD = National Programme on Substance Abuse Deaths.

involves organising codes into categories developed by the research team. The codes are revised and updated as analysis progresses [28]. We chose this method because we had a large number of reports with inconsistent detail; many with just a few words and some with several pages of narrative. The framework analysis meant that each report could contribute to the framework. We also chose to use framework analysis because wanted to explain a phenomenon (hospital-related overdose risk) rather than develop theory.

In developing the framework, we used an iterative deductive and inductive approach. This means a combination of building on a-priori theory (deductive) and identifying themes from the data (inductive). Our deductive work was informed by literature review, existing concepts relating to the 'risk environment' in hospitals [11,29], and discussions amongst the research team before analysis. We developed the framework in five stages: (a) we selected 25 random cases, which three authors (DL, CC, and TB) independently open-coded in triplicate. This

involved adding tags and notes to the free-text coroner reports; (b) we reviewed the codes together and created an initial framework by identifying themes and grouping the themes into categories; (c) we applied the initial framework to all cases, with one author (DL, CC, TB, MH, or KR) coding each case; (d) we reviewed the results collaboratively and revised the framework in several iterative rounds.

We described the thematic framework, including representative quotes from coroners' reports. Although reports are publicly available (for example via Freedom of Information requests), we edited quotes to minimise the chance that an individual might be identified in our report. These edits included assigning a male or female gender at random, using a different name, replacing dates with random dates, and replacing hospital names with fictitious names and locations.

### Ethics and approvals

The King's College London Biomedical & Health Sciences, Dentistry, Medicine and Natural & Mathematical Sciences Research Ethics Sub-Committee re-confirmed (August 2022) that analyses of NPSAD do not require ethics review as all subjects are deceased. In addition, this manuscript has been reviewed by the ethics committee and NPSAD executive committee to ensure that published data and quotes respect confidentiality of decedents.

## Results

### Characteristics of cases

We identified 121 cases that met our inclusion criteria. 42 people died while admitted to hospital and 79 died in the 14 days after discharge. The two groups had similar demographic characteristic: the median age was 40 (IQR 34–46) and 88/121 (73%) were male. The age and sex distributions reflect the characteristics of people who died after using opioids in England [7,25]. The largest proportions were in the North West and Yorkshire and the Humber. The geographical distribution may reflect differences in the prevalence of opioid use and differences in coroners' participation in NPSAD.

70/121 (58%) were admitted to hospital for treatment of medical problems (rather than a drug overdose) and 32/121 (26%) were admitted due to a drug overdose. We could not determine the cause of hospital admission for the remaining cases. A large proportion had other sedatives in addition to opioids identified at post-mortem. Benzodiazepines were particularly common and were detected in over half of toxicology reports. Characteristics of study participants are summarised in Table 1.

### Thematic framework analysis of factors that may contribute to the risk of opioid overdose during or shortly after hospital admissions

We developed a framework consisting of eight themes organised into three categories: (a) hospital actions and policies; (b) high risk use of sedatives; and (c) declining health. Themes and categories are summarised in Table 2.

**(a) Hospital actions and policies.**   Four themes related to hospital actions and policies: concealing drug use or using drugs in unsafe places; self-discharge leading to missed opportunities for care; poor discharge location such as the street or a temporary hostel; and expecting low quality care or the patient bringing their own treatment.

Many reports indicated covert drug use during a hospital admission. These included hospital staff reporting that they had seen patients trying to hide drug use, temporarily leaving a ward and returning intoxicated, and patients found unconscious or dead with evidence of

**Table 1. Characteristics of study participants; data from NPSAD 2010–2021.**

| Variable | Level | Died during hospital admission | Died in the 14 days after hospital discharge | Total |
|---|---|---|---|---|
| Total | | 42 (100.0) | 79 (100.0) | 121 (100.0) |
| Age | 18–24 | 1 (2.4) | 4 (5.1) | 5 (4.1) |
| | 25–34 | 13 (31.0) | 14 (17.7) | 27 (22.3) |
| | 35–44 | 16 (38.1) | 38 (48.1) | 54 (44.6) |
| | 45–54 | 8 (19.0) | 20 (25.3) | 28 (23.1) |
| | 55–64 | 3 (7.1) | 2 (2.5) | 5 (4.1) |
| | 65+ | 1 (2.4) | 1 (1.3) | 2 (1.7) |
| | Median [IQR] | 39 [33–46] | 40 [36–46] | 40 [34–46] |
| Sex | Female | 10 (23.8) | 23 (29.1) | 33 (27.3) |
| | Male | 32 (76.2) | 56 (70.9) | 88 (72.7) |
| Housing status | Housed—alone | 18 (42.9) | 30 (38.0) | 48 (39.7) |
| | Housed—with family or friends | 6 (14.3) | 24 (30.4) | 30 (24.8) |
| | No fixed abode | 1 (2.4) | 8 (10.1) | 9 (7.4) |
| | Not known | 17 (40.5) | 17 (21.5) | 34 (28.1) |
| Region | East Midlands | 1 (2.4) | 1 (1.3) | 2 (1.7) |
| | East of England | 6 (14.3) | 4 (5.1) | 10 (8.3) |
| | London | 1 (2.4) | 2 (2.5) | 3 (2.5) |
| | North East | 2 (4.8) | 1 (1.3) | 3 (2.5) |
| | North West | 7 (16.7) | 30 (38.0) | 37 (30.6) |
| | Northern Ireland | 1 (2.4) | 4 (5.1) | 5 (4.1) |
| | South East | 5 (11.9) | 8 (10.1) | 13 (10.7) |
| | South West | 2 (4.8) | 1 (1.3) | 3 (2.5) |
| | West Midlands | 3 (7.1) | 12 (15.2) | 15 (12.4) |
| | Yorkshire and the Humber | 13 (31.0) | 13 (16.5) | 26 (21.5) |
| | Other | 1 (2.4) | 3 (3.8) | 4 (3.3) |
| Discharge against medical advice | Yes | NA | 28 (35.4) | NA |
| | No | NA | 33 (41.8) | NA |
| | Unknown | NA | 18 (22.8) | NA |
| Admitted due to drug overdose | Yes | 6 (14.3) | 26 (32.9) | 32 (26.4) |
| | No | 27 (64.3) | 43 (54.4) | 70 (57.9) |
| | Unknown | 9 (21.4) | 10 (12.7) | 19 (15.7) |
| Type of hospital | Acute medical | 29 (69.0) | 66 (83.5) | 95 (78.5) |
| | Psychiatric | 13 (31.0) | 9 (11.4) | 22 (18.2) |
| | Unknown | 0 (0.0) | 4 (5.1) | 4 (3.3) |
| Classes of sedatives found at post-mortem (in addition to opioids) | Antidepressants | 11 (26.2) | 29 (36.7) | 40 (33.1) |
| | Antipsychotics | 7 (16.7) | 13 (16.5) | 20 (16.5) |
| | Benzodiazepines | 22 (52.4) | 48 (60.8) | 70 (57.9) |
| | Gabapentinoids | 6 (14.3) | 25 (31.6) | 31 (25.6) |
| | Z-drugs | 5 (11.9) | 12 (15.2) | 17 (14.0) |
| | At least one | 27 (64.3) | 61 (77.2) | 88 (72.7) |
| Classes of sedatives prescribed (other than opioids) | Antidepressants | 11 (26.2) | 29 (36.7) | 40 (33.1) |
| | Antipsychotics | 8 (19.0) | 12 (15.2) | 20 (16.5) |
| | Benzodiazepines | 8 (19.0) | 17 (21.5) | 25 (20.7) |
| | Gabapentinoids | 6 (14.3) | 15 (19.0) | 21 (17.4) |
| | Z-drugs | 3 (7.1) | 12 (15.2) | 15 (12.4) |
| | At least one | 16 (38.1) | 43 (54.4) | 59 (48.8) |

**Table 2. Themes relating to the risk of opioid-related deaths during hospital admissions or shortly after hospital discharge.**

| Category | Themes | Number of cases (n = 121) | Example case |
|---|---|---|---|
| (a) Hospital actions and policies | (1) Concealing drug use, or using drugs in unsafe places | 23 | Admitted to hospital due to stomach pain. Medical staff were aware of drug and alcohol use. Five days after admission the patient was discovered in a hospital bathroom with drug paraphernalia [ID111]. |
| | (2) Discharge against medical advice leading to missed opportunities for care | 30* | Admitted to hospital after a heroin overdose and naloxone was administered. Left hospital against medical advice later that evening, and was found dead at a relative's house after using heroin and benzodiazepines [ID17]. |
| | (3) Poor discharge location, such as the street or a temporary hostel | 16 | Admitted to hospital after a suspected overdose and being found unconscious in a public place. Admitted to hospital for three weeks, and then discharged to a hotel. Did not respond to knocking on the day of check-out from the hotel [ID60]. |
| | (4) Expecting low quality care or bringing own treatment | 11 | Admitted for treatment of pneumonia. Brought additional, non-prescribed methadone into hospital and died after using this methadone [ID51]. |
| (b) High risk use of sedatives | (5) Increased use of sedatives due to acute illness | 28 | Had a fall and was admitted to hospital with a fractured wrist and dislocated shoulder. Was discharged, and then died after taking additional opiates and antidepressants to manage pain [ID45]. |
| | (6) Loss of tolerance to opioids | 35 | Medical records of using heroin, crack cocaine, alcohol, and had a methadone prescription. Admitted to hospital due to pneumonia and detoxed during the admission. Died the day after discharge [ID46]. |
| (c) Declining health | (7) Poor health or mobility reducing access to treatment for substance use | 9 | History of heroin use. Had COPD and cellulitis on legs and feet, which severely affected mobility and may have prevented access to community drug treatment services. Admitted to hospital with shortness of breath, but self-discharged. Later died at home after using heroin. Post-mortem toxicology suggested methadone was not recently used, despite having a methadone prescription [ID6]. |
| | (8) Acute exacerbations of long-term conditions or sudden deteriorations in health, increasing vulnerability to fatal overdose | 51 | Medical history of asthma and heroin use. Admitted to hospital with respiratory failure and treated for asthma exacerbation. Self-discharged the day after admission. Went to bed that evening with shortness of breath and died after using heroin [ID90]. |

* In two reports where the death occurred during a hospital admission, discharge against medical advice was noted for a previous admission, hence 30 reports have discharge against medical advice in this table, compared to 28 in Table 1.

drug use, such as a syringe or "heroin powder found at the scene in clothes pocket" [ID82]. One patient was admitted due to an infected injecting wound in the groin, and had recently returned to the ward after a scan:

> "At approximately 0417hrs a nurse witnessed Paul to be fumbling around with something in a pouch and doing something around his groin area. A short while later, a healthcare assistant noticed he looked a lot more unwell, also noticed Paul was bleeding from the right of his groin again. Paul was losing a lot of blood" [ID10].

Some reports stated that patients had community-based prescriptions of OAT, but it was unclear if OAT was continued in hospital. In one report, a person who used heroin and had a methadone prescription was admitted to hospital due to a pulmonary embolism and was "believed to have been using heroin (by injection) whilst being treated" [ID117]. The report did not mention a methadone prescription in hospital. Similarly, another patient had an infection and was admitted to hospital for intravenous antibiotics. The coroner's report stated that they used heroin while in hospital. Although they had a long history of drug and alcohol problems and a prescription of methadone prior to admission, methadone was not identified in

post-mortem toxicology testing which may suggest it was not provided in hospital [ID83]. Insufficient OAT or opioid withdrawal management may have led to illicit drug use, and some patients underwent opioid 'detox' (withdrawal management) in hospital without initiation of OAT, which is known to increase risks of fatal overdose.

Many reports included accounts of discharge against medical advice. These accounts typically use language such as "absconded" or "failure to comply". One patient who had endocarditis (a serious infection of the heart) was described as "constantly disappearing off the ward for cigarettes" and that they "failed to comply with [the] hospital". Two days after admission, this patient left hospital, and was later found unresponsive at a relative's house [ID52]. Many accounts of discharge against medical advice suggested drug use very shortly after leaving hospital, and often in risky places. For example, one patient left hospital after being admitted due to self-harm, and immediately took a fatal overdose:

> "The deceased had a past medical history of alcohol and substance misuse and deliberate self-harm. He was admitted to King George V Hospital on 7th February 2013 with self-inflicted wounds to the wrist. He also requested assistance with detoxing and he showed signs of withdrawal. A detox regime was commenced. On 12th February 2013, he absconded from the ward and was readmitted following a cardiac arrest. He was said to have taken cocaine, heroin, and alcohol, and collapsed outside a friend's room" [ID9].

Both for patients who left hospital against medical advice, and for those who completed treatment, the location of discharge was often poor. After leaving hospital, patients in our sample were recorded as sleeping in locations such as temporary hostels, hotels, private vehicles, tents, or on the street. Many used drugs in an unsafe or unfamiliar place while unwell, and reports included accounts of patients found dead in these locations. One patient was discharged from hospital after an overdose of benzodiazepines and antidepressants. This patient was discharged to supported accommodation, but the coroner's report suggests that staff were not present that night and were unaware of this recent overdose. The patient later died from an overdose of heroin and other sedatives [ID1].

In some reports, patients brought their own drugs into hospital, which may reflect expectations that opioid withdrawal or pain would not be managed. One patient was admitted with gastrointestinal issues and liver cirrhosis. After they were found unconscious, "staff found an empty methadone bottle in her name and some brown and white tablets not prescribed by the hospital" [ID49]. In another case, a patient was admitted to hospital for a respiratory infection and had a medical history including obstructive hiatus hernia, sleep apnoea, obesity, and alcohol and cocaine use. During the hospital admission this patient told hospital staff that they used morphine prescribed to a family member. This patient was discharged after one day in hospital and found dead three days later with morphine contributing to death [ID13].

**(b) High risk use of sedatives.** Many of the coroner reports included evidence that people who died took multiple sedatives or drank alcohol in addition to using opioids. Out of the cases included in our sample, 59/121 (49%) had prescriptions of multiple sedatives and 88/121 (73%) had another sedative in addition to opioids detected at post-mortem, most commonly benzodiazepines (Table 1). While use of sedatives may not be unusual among people who use illicit drugs, some reports suggested increased use of sedatives to manage symptoms of acute illness, pain, mental health crises, or alcohol withdrawal. For example, a patient was admitted to hospital for 13 days due to severe back pain. The patient's medical notes included use of heroin, crack cocaine, and dependence on unspecified prescription drugs. On discharge, the patient was given a "large amount" of take-home morphine, diazepam, and pregabalin. They were discharged to temporary accommodation and died three days later due to a drug

overdose [ID99]. In several cases, a mental health crisis appears to have caused hospital admission and risky sedative use. For example, one individual attended hospital after taking several sedatives. They left hospital before being assessed, and later died from an overdose attributed to opioids, pregabalin, and benzodiazepines:

> "The deceased had a history of alcoholism, polysubstance abuse, dissocial personality traits, depression and schizophrenia. She was taken from Queen Mary Street to the Accident and Emergency Department of the University Hospital Manchester, by ambulance at around 10.15am on the 4th July 2012. When triaged, she complained of low mood thoughts of self-harm, visual and auditory hallucinations. She stated that she had taken Lyrica tablets, Diazepam and Xanax tablets the previous day . . . A preliminary mental health assessment was started but a full assessment could not be completed as the deceased was under the influence of drugs. As a result she was admitted in order to allow her time to become medically fit for the assessment to take place but the deceased left the hospital" [ID108].

In many cases multiple sedatives were present in toxicology reports, but we could not determine whether using sedatives in that way was unusual for a given individual. For example, one individual had a history of heroin use and prescriptions including methadone, morphine, pregabalin, mirtazapine, and quetiapine. They were admitted to hospital for treatment of pneumonia. This patient died on the day after discharge after using heroin, diazepam, and drugs prescribed during the hospital admission [ID3]. In cases such as this, symptoms of an acute illness may have led individuals to use higher doses of prescribed or illicit drugs for symptom management. Toxicology reports also commonly identified unregulated and non-medical 'street' benzodiazepines such as etizolam and flualprazolam. We were unable to determine if use of these 'street' benzodiazepines was established before the time hospital admission or was initiated following discharge.

Hospital admissions may also be associated with loss of tolerance to opioids and other sedatives. This may be due to periods of abstention while in hospital because drugs are unavailable, intentional 'detox' in hospital, or insufficient opioid withdrawal management or OAT. Some of these issues are explored in the themes under 'hospital actions and policies', above. This appears particularly important for patients in psychiatric hospitals who may have long stays or more limited access to illicit drugs. One patient was an involuntary inpatient being treated for psychosis for eight weeks. They left the hospital and died the next day after using heroin [ID22]. Another patient was detained for ten days in a psychiatric hospital died shortly after release:

> "The deceased was a long-standing known drug user and alcoholic . . . He had recently been detained between the 8th to the 18th November 2021 at the Ivy Ward and it is possible he had lost a level of tolerance to the drugs he had been injecting or consuming. Postmortem samples demonstrated the deceased had recently used heroin, methadone, pregabalin, cocaine, paracetamol, mirtazapine, amitriptyline, quetiapine, diazepam, and omeprazole" [ID78].

The coroners' reports often showed that hospital staff were aware of patients' high risk sedative use, which may suggest there were opportunities to mitigate this risk, for example through initiation or continuation of OAT or provision of take-home naloxone. From individual reports, we were usually not able to determine whether hospitals attempted to mitigate the risk of overdose.

**(c) Declining health.** Most hospital admissions were for treatment of physical health problems and exacerbations of long-term conditions, rather than drug-related problems such as overdoses and intoxication. Many patients had poor respiratory health or bacterial infections. In some reports these physical health problems meant that patients were unable to access community support after discharge. In one case, hospital staff identified poor conditions in a patient's home and the patient's difficulty in attending a community drug treatment service due to mobility problems:

"James was an intravenous drug user who used to inject in his groin. On 17th November 2017, he was admitted to hospital with a two-day history of sudden-onset pain in his left groin . . . accompanied by feeling weak and feverish . . . He had stopped methadone treatment a month before. On 19th November, he underwent surgical exploration of the groin, at which a large pseudoaneurysm was revealed . . . By 5th December 2017, James was deemed fit to be discharged with a dressing. On 9th December, he was reviewed in hospital to assess his current health and social issues: several problems were identified. These included the insanitary condition of James's flat, his difficulties accessing methadone maintenance services owing to limitation by pain of his ability to walk to the clinic" [ID110].

Health-related barriers to healthcare were evident in many reports. Another patient had long-term problems with drugs and respiratory health, but limited support for these problems. This patient was admitted to hospital for treatment of an acute exacerbation of chronic obstructive pulmonary disease (COPD), was discharged home, and died six days later after using heroin. A General Practitioner report stated that:

"Andrew had a longstanding alcohol problem and heroin addiction, dating back at least 20 years and diagnosed with COPD 11 years ago. . . Andrew was admitted to Hastings Hospital six times in the year prior to his death, with chest infection and respiratory failure . . . On 2 May 2018 I had a telephone consultation with Community Matron regarding low oxygen saturations . . . Andrew was advised to have admission to hospital immediately, but the patient refused . . . He felt lonely and helpless but declined hospitalisation . . . we agreed that Andrew was a Palliative Care case now. The patient was admitted to hospital on 4 May 2018 by 999 ambulance after self-referral and discharged on 9 May 2018 [with a diagnosis of infective exacerbation of COPD]. The patient then attended by ambulance on 15 May 2018 having been found by his mother . . . Unfortunately, the patient had poor history of attending outpatient appointments, which were arranged for him–both endocrinology team and respiratory medicine . . . I am not aware of a Drug and Alcohol Team being involved with this patient's care for some years [ID105].

Some coroner reports similarly suggested that patients might have reduced mobility, for example due to severe COPD or a long hospital admission, and documented a lack of opioid agonist therapy, though the link between mobility and drug treatment was not explicit. One patient with COPD and mobility problems due to soft tissue infections made multiple unsuccessful attempts to be assessed for shortness of breath. Although this patient had a methadone prescription, methadone was not identified in post-mortem toxicology testing:

"Michelle had a previous medical history of cellulitis of the foot, drug addiction therapy (methadone), and COPD. Michelle had a makeshift bed in the kitchen due to her poor mobility as she had abscesses on both of her feet and legs, she was in ill health and had been admitted to hospital recently (twice in one week). On 12th January Michelle's partner had

called an ambulance due to her being short of breath. Paramedics attended and conducted basic health checks but Michelle refused to go with them. Michelle was later taken to hospital with a friend but as she was left for approximately 12 hours without being seen, she got fed up and left. The following day Michelle's partner called emergency services again as her breathing was getting worse. Michelle went with the ambulance to hospital where she only stayed for a few hours as she stated that she was left again" [ID6].

Many coroner reports described acute exacerbations of long-term conditions or other sudden deteriorations of health, which may have led to increased vulnerability to fatal overdose. These reports most frequently described acute respiratory health problems such as pneumonias and other acute exacerbations of COPD. For example, one report described a patient who died six days after being discharged for treatment of pneumonia:

"Discharged from hospital 9 August 2014 having suffered pneumonia and sepsis. On 15 August 2014 he was struggling to breathe. Ambulance called, there was a [do not attempt cardiopulmonary resuscitation] order in place, and he was declared deceased by paramedics. History of heroin, crack, and cannabis use, HIV, COPD with infective exacerbations" [ID113].

In several cases including the one above, the cause of death was listed as a combination of opiates or other sedatives and COPD. Several coroners' reports described decompensations of liver disease, often with comorbidities, which may have contributed to the risk of opioid overdose through impaired metabolism, encephalopathy, or frailty. One patient had a history of alcohol-related liver cirrhosis, chronic pancreatitis, and secondary diabetes. They were admitted to hospital due to "severe pancreatitis" and died after taking methadone in hospital [ID89]. In this case, it was unclear whether methadone was prescribed.

Some reports described multiple health problems, very severe health problems, or patients who were near the end of life. In these cases, the use of palliative care pathways or the availability of specialist palliative care services was unclear from the coroners' reports.

## Discussion

### Key findings

In this qualitative analysis of coroner's reports, we identified several factors potentially contributing to excess risk of opioid-related deaths soon after hospital discharge. These include the risk of using drugs in a hospital setting, such as using drugs in locked bathrooms; high-risk use of sedatives due to acute illnesses and loss of opioid tolerance; and the effect of poor health on respiratory depression and access to community drug treatment services. The coroners' reports show that patients' use of drugs was often known to hospital staff. Deaths may be prevented by clearer policies to help these patients remain in hospital, complete treatment, and discharge with a plan to access stable housing and services such as OAT while recovering.

### Comparisons with previous studies

We are not aware of previous research that attempts to explain the excess risk of fatal overdose after hospital discharge. However, other research has sought to explain why risk increases at other critical moments. One such critical moment is release from prison, which has a well-known association with fatal drug poisoning [1,2]. One explanatory model [30] includes four pathways: (a) 'underlying factors' including opioid use among people in prison, limited access to OAT, health states including pain and mental health problems, and trauma; (b)

'intermediate factors' including disrupted social networks, interruptions in care, and stigma; (c) 'proximate factors' including solitary use, insufficient naloxone access, and poly-drug use; and (4) 'biological factors' including reduced opioid tolerance. Hospital admissions have many differences to prison stays, including the typically shorter duration (the mean custodial sentence for drug offenses in England and Wales was 40 months in 2020 [31]; compared to 9 days for hospital admissions prior to drug-related deaths [7]), that all hospital patients are unwell, and most hospital admissions are voluntary. However, the factors that may explain the large number of deaths after prison release also plausibly explain deaths after hospital discharge; though some factors are likely to be more important, such as health states.

Another critical moment is cessation of OAT, which may confer risk through loss of tolerance to opioids [3]. For example, a study in England showed that mortality rates during the first two weeks after discharge from methadone or buprenorphine treatment was nine times that of patients on treatment [32].

There may be common processes that increase risk across these critical moments. A study of non-fatal overdoses in Canada found several such critical moments, including hospital discharge, entry to prison, release from prison, new prescriptions of sedatives, discontinuing antipsychotics, and exiting methadone treatment [10]. These events are associated with unusual drug use, changing tolerance, stressful life events, and disruptions to treatment and support services. The authors of this study concluded that better continuity of care, particularly OAT, during periods of transition could prevent overdoses.

## Relevance for policy and practice

The practical implications of this study relate to acute hospitals and how they support patients who use illicit opioids. Recent reviews of hospital policies in England [16] and the United States [17] showed that many require disproportionate caution in prescribing OAT, such as requiring urine screens or slow titration. In addition, qualitative research has shown that many hospital staff do not feel they have sufficient knowledge to treat patients who are dependent on opioids [19,33]. As a result, patients who use opioids may delay or interrupt hospital care [18]. Our results suggest that hospitals may be able to provide better support in five areas.

First, providing sufficient OAT during periods of acute illness. The accounts we reviewed show that hospital patients may use drugs covertly, and OAT may help prevent this. Basic training on substance use and OAT could help medical staff initiate and continue OAT for patients during hospital treatment. After discharge, patients may have reduced mobility and be unable to attend community pharmacies or OAT clinics, and therefore use illicit drug dealers who are willing to visit clients' homes. Some cases in our study were unable to access OAT due to illness and died after using illicit drugs. In these cases, hospital staff may need to work with community drug services to arrange home visits and delivery of medication.

Second, providing harm reduction interventions that recognise illicit drug use is likely to happen. Naloxone may be particularly beneficial when a patient comes to hospital with a friend or family member. Hospitals have piloted take-home naloxone, particularly in emergency departments [34,35], though this is uncommon in part due to lack of understanding of naloxone [34]. Tolerated and managed use of illicit drugs may prevent the riskiest situations such as patients using drugs alone in locked bathrooms and toilet cubicles, which was common in our study. Supervised consumption rooms have been set up in some hospitals in Canada [36,37] and evaluations suggest they are safe and improve medical care, including by prevention of early discharge due to opioid withdrawal [38].

Third, improving discharge planning for patients who use drugs. In the coroners' reports in our study, many discharges appeared sudden and unsupported, with patients leaving

unilaterally or going to an unsafe location. Further research into the clinical circumstances of 'discharge against medical advice' may inform potential interventions. For example, staff may have opportunities to discuss patients' needs or provide pre-prepared packs including naloxone and emergency contacts. Even when patients are considered ready for discharge, some do not have a suitable discharge location. Our study included accounts of patients using drugs in cars, public toilets, and on the street after hospital discharge. There is substantial pressure on acute hospitals to discharge patients and free up beds [39]. Specialist step-down care (or accommodation with clinical support, also known as 'medical respite' or 'intermediate' care) for people who are homeless or use drugs may relieve pressure on hospitals and help staff arrange safe discharges [40–42]. A pilot of specialist step-down care for hospital patients who would otherwise be discharged to street in England found that it reduced delayed transfers of care (sometimes known as "bed blocking") and emergency readmissions [43].

Fourth, identifying 'poly-drug' use and reviewing medicines. Many people who use illicit opioids also use other drugs, such as cocaine, benzodiazepines, gabapentinoids, and alcohol [44,45]. Simultaneous use of multiple sedatives is particularly risky. Poly-drug use has increased over the past decade and an increasing proportion of opioid-related deaths in the UK and North America have multiple substances identified at post-mortem [46,47]. In many reports that we reviewed, patients' poly-drug use or multiple sedative prescriptions were noted in medical records. At the time of hospital admission and acute illness, the need for pain relief may be increased and patients may use increased doses of prescription or illicit sedatives. These risks can be complex to mitigate and limiting prescriptions of these medicines could deny patients pain relief and increase use of more dangerous illicit alternatives (such as the unregulated 'street' benzodiazepines that are particularly common in Scotland [48]). Where clinicians know that patients use multiple sedatives, they may be able to reduce risk by discussing the patient's needs, if necessary, with advice from pain and addiction specialists, and ensuring that patients understand how to use medicines and can be monitored by friends, relatives, or healthcare professionals.

Fifth, involving palliative care teams. Some individuals in our study had very severe illnesses and some deaths are not avoidable. However, the manner of death was often poor. Social exclusion is associated with poor access to palliative care; for example, qualitative research has shown that barriers to palliative care include stigma, insecure accommodation, and limited capacity among hostel staff [49].

## Strengths and limitations

We used a unique database of coroners' reports to explore the circumstances of deaths due to drug poisoning during and shortly after hospital admissions, which allowed us to discuss reasons for the high number of such deaths. The study has three key limitations:

First, NPSAD is an incomplete database. This is partly due to varying coverage, with some coroners not submitting reports. Coverage varies by region, which is likely to explain the small number of reports in some regions (Table 1). In addition, many reports include limited detail, meaning we could not determine whether the case met inclusion criteria. Therefore, the number of cases included in our study does not reflect the actual number of people who died in relevant circumstances. Our recent quantitative study [7] covering a similar period estimated 1,088 opioid-related deaths in the two weeks after hospital discharge, compared to 78 in this study, suggesting that we captured approximately 7% of cases. We believe that the most important selection bias is likely to occur due to coroners' decisions to include more detailed reports for certain cases. It is possible that coroners might include more detail where the actions of health services may have contributed to a death and there may be questions relating to

professional practice or neglect. Conversely, coroners may include less detail where a death is attributed to 'natural causes'. Therefore, our sample may be more likely to include cases with themes in category A ('hospital actions and policies), and less likely to include cases with themes in category C ('declining health').

Second, coroners' reports included varying and limited data. We interpreted the available data, which was sometimes detailed and sometimes sparse. We were unable to judge the quality of clinical care based on these accounts. For example, we could not judge the appropriateness of sedative prescribing, or whether hospital staff could have prevented early discharge.

Third, the thematic framework and conclusions are dependent on the researchers' interpretations of the data. We tried to maximise reliability between researchers by creating a coding framework together based on a sample of reports, and iteratively revising the framework together. We recognise that we have prior beliefs that affect this process, particularly our support for 'harm reduction' interventions that aim to increase the safety of illicit drug use rather than stop it completely. This approach might be contrasted with abstinence, zero-tolerance, or recovery-focused approaches that aim to prevent illicit drug use. Researchers with prior beliefs in these alternative paradigms might have reached different conclusions, particularly regarding tolerance of drug use in hospitals.

## Conclusion

Hospital discharge is associated with increased risk of death due to illicit opioid use. Patients who use illicit opioids may leave hospital to use drugs or use drugs covertly while in hospital. Hospitals need guidance to help them care for patients who use illicit opioids, particularly in relation to improving withdrawal management, harm reduction interventions such as take-home naloxone, discharge planning including continuation of opioid agonist therapy during recovery, management of poly-sedative use, and access to palliative care.

## Author Contributions

**Conceptualization:** Dan Lewer, Thomas D. Brothers, Magdalena Harris, Kirsten L. Rock, Caroline S. Copeland.

**Data curation:** Caroline S. Copeland.

**Formal analysis:** Dan Lewer, Thomas D. Brothers, Magdalena Harris, Kirsten L. Rock, Caroline S. Copeland.

**Investigation:** Dan Lewer, Thomas D. Brothers, Magdalena Harris, Kirsten L. Rock, Caroline S. Copeland.

**Methodology:** Dan Lewer, Thomas D. Brothers, Magdalena Harris, Kirsten L. Rock, Caroline S. Copeland.

**Writing – original draft:** Dan Lewer, Thomas D. Brothers, Magdalena Harris, Kirsten L. Rock, Caroline S. Copeland.

**Writing – review & editing:** Dan Lewer, Thomas D. Brothers, Magdalena Harris, Kirsten L. Rock, Caroline S. Copeland.

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
