## [Decision Letter · Decision Letter 0]

6 Jan 2023

PONE-D-22-27646Opioid-related deaths during hospital admissions or shortly after discharge in the United Kingdom: a thematic framework analysis of coroner reportsPLOS ONE

Dear Dr. Lewer,

Thank you for submitting your manuscript to PLOS ONE. After careful consideration, we feel that it has merit but does not fully meet PLOS ONE’s publication criteria as it currently stands. Therefore, we invite you to submit a revised version of the manuscript that addresses the points raised during the review process. We have received one review for this manuscript, however finding additional reviewers has been a challenge.  So as not to delay further, as Academic Editor, I have also undertaken a review.  The reviewer recommended that the manuscript be rejected.  However, there is significant value in this work despite the limitations and potential for bias in the source data.  The manuscript was well written and well presented.  Please address the reviewers comments (at the bottom of this email), but focusing specifically on the following:Addressing the balance in the background between objective 1 & 2.  Given the issues with the data you may which to consider removing objective 1 and focus on the themes from the reports.  Amend the background to make it clear what is evidenced in research and what are the authors thoughts e.g. this wording 'Potential mechanisms behind the increased risk after hospital discharge have not been studied, but could include' - Also I would recommend removing the reference to newspaper reports and just include academic research evidence.Please include in the background a discussion and rationale for using thematic framework analysis to documents.Please add to the limitations sections issues around bias. Please submit your revised manuscript by Feb 20 2023 11:59PM. If you will need more time than this to complete your revisions, please reply to this message or contact the journal office at plosone@plos.org. Please include the following items when submitting your revised manuscript:A rebuttal letter that responds to each point raised by the academic editor and reviewer(s). You should upload this letter as a separate file labeled 'Response to Reviewers'.A marked-up copy of your manuscript that highlights changes made to the original version. You should upload this as a separate file labeled 'Revised Manuscript with Track Changes'.An unmarked version of your revised paper without tracked changes. You should upload this as a separate file labeled 'Manuscript'.

We look forward to receiving your revised manuscript.

Kind regards,

Charlotte Lennox

Academic Editor

PLOS ONE

Journal Requirements:

No

Reviewers' comments:

Reviewer's Responses to Questions

**Comments to the Author**

1. Is the manuscript technically sound, and do the data support the conclusions?

Reviewer #1: Partly

2. Has the statistical analysis been performed appropriately and rigorously? 

Reviewer #1: N/A

3. Have the authors made all data underlying the findings in their manuscript fully available?

Reviewer #1: No

4. Is the manuscript presented in an intelligible fashion and written in standard English?

Reviewer #1: Yes

5. Review Comments to the Author

Reviewer #1: This study addresses a topic of high importance and relevance, examining the potential mechanisms underlying the increased risk of illicit drug overdose deaths following hospital discharge. Research is needed to help inform strategies for overdose prevention and engagement in harm reduction and other forms of care. However, the novel contribution of this study is unclear, given limitations in the data/approach.

The objectives of the study are to describe the characteristics of patients who died due to an opioid overdose during or shortly after discharge, and to identify the contributing factors. With respect to the second objective, the authors note that prior studies tend to rely on hospital data that offer limited contextual information with which to identify factors that contribute to fatal overdoses. However, several potential contributing factors are outlined in the Background/literature review, including some backed with citations of prior research. It isn’t clear whether these factors were identified empirically in these prior studies or through speculation, clinical experience, or other means. Further, no attention is given in the Background to establishing a rationale for the first objective. Without this information, the rationale and novel contribution of the current study (which identifies many of these same factors) are not clearly established.

The data source has limitations that may restrict the impact of this work beyond what is acknowledged in the Discussion. The limited coverage of fatal overdoses, in combination with the voluntary nature of reporting, differences in levels of detail, and unknown (but likely) reporting biases within and across geographical areas and/or coroners, all detract from data quality. Given questions of representativeness, it is not clear that these data are suited to answer the first objective (providing a description of patient characteristics). Some variables in Table 1 have a high level of missing values, hindering interpretation. Further, unknown biases affected what is and isn’t recorded across coroners may have impacted the identification of contributing factors. In describing study limitations, the authors note the relatively low coverage of fatal overdoses in the data source but that they do not think there’s reason to suspect bias. This seems insufficient justification.

Additional more minor comments are as follows. Use of the risk environments concept fits for this study, but more detail would be appreciated on how this framework was used in the analysis (e.g., what dimensions of context were considered and whether these were limited to hospital contexts). It seems like attention to community contexts may have also been informative, such as what services are available, what is known about the illicit drug supply, its composition and potential variation across communities. Although it makes sense that focus would be placed on hospital contexts, community contexts will also have affected overdoses as well as what gets potential recorded in coroners’ reports – was this considered?

In table 2, the link between the example case and theme isn’t always clear; alternative explanations seem possible without more information/context. For example, under 7) poor health leading to reduced access to treatment – is the lack of evidence of recent methadone use (even in the context of poor/worsening health) sufficient to suggest access to treatment was impacted by poor health? This seems speculative, and there are other potential explanations for why OAT was not evident. Perhaps the case description is not fully indicative of the information contained in the report, meaning there were other reasons to attribute this death to a lack of access to treatment. It just isn’t clear in what is provided, and so raises questions about how attributions were made during analysis.

In the context of the toxic and volatile illicit drug supply, the loss of opportunities for engaging people in potentially life-saving harm reduction and health care during hospital and other service encounters is a major issue that needs to be remedied. The quotes in this paper are terribly sad and serve to highlight important shortcomings in a way that affects the reader. However, these shortcomings appear to have been demonstrated to some degree previously; when combined with the methodological limitations of this study, its overall contribution to the literature appears limited.

6. PLOS authors have the option to publish the peer review history of their article (what does this mean?). If published, this will include your full peer review and any attached files.

Reviewer #1: No

---

## [Author Response · Author response to Decision Letter 0]

2 Mar 2023

Please see attached document, including point-by-point responses.

---

## [Editor Report · Decision Letter 1]

13 Mar 2023

Opioid-related deaths during hospital admissions or shortly after discharge in the United Kingdom: a thematic framework analysis of coroner reports

PONE-D-22-27646R1

Dear Dr. Lewer,

We’re pleased to inform you that your manuscript has been judged scientifically suitable for publication and will be formally accepted for publication once it meets all outstanding technical requirements.

Kind regards,

Charlotte Lennox

Academic Editor

PLOS ONE
---

## [Editor Report · Acceptance letter]

11 Apr 2023

PONE-D-22-27646R1 

Opioid-related deaths during hospital admissions or shortly after discharge in the United Kingdom: a thematic framework analysis of coroner reports 

Dear Dr. Lewer:

I'm pleased to inform you that your manuscript has been deemed suitable for publication in PLOS ONE. Congratulations! Your manuscript is now with our production department. 

Kind regards, 

on behalf of

Dr. Charlotte Lennox 

Academic Editor

PLOS ONE